# Full Waveform Prediction of Blasting Vibration Using Deep Learning

**Yunsen Wang \*, Guiping Zheng, Yuanhui Li and Fengpeng Zhang**

Key Laboratory of Ministry of Education on Safe Mining of Deep Metal Mines, Northeastern University, Shenyang 110819, China; zhengguiping@mail.neu.edu.cn (G.Z.); liyuanhui@mail.neu.edu.cn (Y.L.); zhangfengpeng@mail.neu.edu.cn (F.Z.)
\* Correspondence: wangyunsen@mail.neu.edu.cn

**Abstract:** Blasting vibration could cause dynamic instability of rock masses within a critical steady state. To control the blasting vibration, it is necessary to understand the complete dynamic response process of the rock masses under the blasting vibration. The Long Short-Term Memory (LSTM) technique uses blast monitoring data to predict the full waveform of the blast vibration. Based on the LSTM, a new full waveform prediction model is proposed in this study. To verify the feasibility of the proposed model, the sample data were constructed using the well-known linear blast wave superposition prediction formula. The full waveform prediction model is trained and the predicted waveform and the actual waveform are then evaluated and compared. The loss function is calculated and discussed, which verifies the feasibility of the prediction method. In addition to the numerical research, the actual blasting vibration data are also used for verification. The parameters, such as sequence size, training algorithm, and some hidden layer nodes, are discussed and optimized. The results show that the proposed full waveform prediction model based on LSTM can predict the full blasting waveform. This study provides a new idea for the prediction and control of blasting vibration.

**Keywords:** blasting vibration; waveform prediction; Long Short-Term Memory

## 1. Introduction

Blasting technology has been widely used in engineering projects such as water conservancy, transportation, construction, and mining. Currently, blasting is one of the most commonly used techniques in mining or stone excavation projects [1–3]. The development of engineering blasting technology has greatly reduced the labor intensity, accelerated the speed of project construction, and improved work efficiency; however, the impact of blasting on ground vibration, flying stones, air shock waves, noise, and dust on the surrounding environment has also become increasingly prominent. The influence of blasting vibration is mainly manifested in buildings in terms of the damage, cracks, collapsed slopes, and frightened humans and animals [4–6]. The ground vibration effect of blasting is a prime pollution hazard of engineering blasting [7,8]. With the complication of the blasting environment and the enhancement of popular awareness of environmental protection, blasting vibration research has become a major research topic in blasting engineering [9–16]; however, blasting vibration effects are complicated and there are many influencing factors. To be best of the authors' knowledge, no perfect theory is suitable for blasting vibration prediction.

In current engineering applications, the prediction of blasting vibration is mainly conducted using an empirical relation [17]. This empirical equation considers the amount of explosives and the distance from the blasting center to the measurement point as the main influencing factors to predict the vibration speed or acceleration of the particle. This equation is only suitable for flat terrain and there are drawbacks in the practical application when using the empirical equation. For non-flat terrain, many studies have proposed a series of elevation correction equations and a particle peak vibration prediction equation [18–21]. There are many influencing factors that could influence the blasting vibration,

such as the explosive type, explosive characteristics, borehole diameter, charging structure, and rock characteristics [22]; however, the relationship between blasting parameters and blasting vibration is complex, with uncertainty and inaccuracy, and it is non-linear. It is difficult for the prediction equation to represent the relationships among the parameters [23]. Therefore, intelligent algorithms [24–26], such as particle swarm algorithms [27,28] and BP neural networks [29–32], are also used to predict the peak velocity and frequency of blasting vibration.

This research has promoted the development of blasting vibration prediction technology to a certain extent and deepened the understanding of blasting vibration; however, the empirical coefficient method endures, although it can only be used to predict the peak value of blasting vibration but not the duration of blasting vibration and the distribution range of its vibration frequency. Under the blasting vibration, the vibration frequency has a great influence on the destruction of nearby buildings. If the main vibration frequency of the blasting vibration is close to the natural frequency of the building, a larger vibration will be generated due to the resonance effect, which could be destructive. In addition, the damage to the structure caused by the blasting vibration duration cannot be ignored. As the vibration duration increases, the harm to the buildings increases. Therefore, the effects of the three factors, which include amplitude, frequency, and duration, need to be considered in the blasting vibration control. Blasting vibration evaluation requires an understanding of the full vibration process [33].

Research on the prediction of the full waveform of blasting vibration started in the 1980s. Hinze [34] proposed a linear superposition model based on the measured single-hole blasting vibration to simulate the vibration signal of porous blasting under different time delay conditions. Blair [35] proposed two non-linear superposition models based on the linear superposition model and compared them with the linear superposition model. In recent years, some scholars have proposed some new blasting vibration prediction models based on the response spectrum and wavelet packet analysis [36–40].

Deep learning has achieved great success in the fields of speech recognition, natural language processing, machine vision, and multimedia [41–43]. Deep learning is not a new method; it is a collective term for a class of learning methods based on the Deep Neural Network (DNN). The DNN originates from the traditional neural network (NN), but shallow NN capabilities are limited, and it has no advantage over other machine learning algorithms. The DNN model is difficult to use in practical engineering due to various constraints at the time and its shortcomings. Therefore, scholars have mostly abandoned DNN to support research into machine learning methods such as support vector machine (SVM) and boosting. Machine learning models such as SVM can be viewed as shallow NN models with only one or two hidden layers. When these models are used, on the one hand, corresponding expensive feature extraction needs to be performed according to different tasks; on the other hand, more copious data cannot be effectively used. However, DNN can make better use of big data with automatic extraction features, and it is readily transferable between tasks. In 1997, the Long Short-Term Memory (LSTM) was proposed to improve the traditional RNN model [44]. The experimental results showed that the LSTM can effectively solve the problem of gradient explosion and gradient disappearance during RNN training, so that the RNN can best use long-range sequence information. Among current multiple deep learning models, the LSTM is particularly suitable for processing sequence data, due to its ability to use long-range dependent information in sequence data.

A method of predicting the complete vibration characteristics of blasting vibration signals in three-dimensional space is proposed based on LSTM deep learning. This model first pre-processes the original monitoring vibration wave, then vectorizes the pre-processed data, groups the learning samples, and then sends them to the LSTM model for training. The trained model is used to predict the full waveform of the monitoring points. In addition, a variety of the latest deep learning technologies are also tested in this study, including the latest models and training techniques in the process of system implementation. These

technologies are briefly introduced and experimentally analyzed. This study provides a reference for blasting vibration propagation, vibration control and prediction.

## 2. Methodology

Prediction of the full waveform of blasting vibration has spatial–temporal complexity. The waveform prediction for the next moment is based on the current state and the previous waveform, including the interaction of the blasting wave between the propagation path media. In this study, three components of the spatial state of the blasting vibration wave are integrated into the LSTM network to obtain a reliable prediction result. The proposed model uses existing technologies, including vibration monitoring, RNN and LSTM, and model training techniques.

### 2.1. LSTM Network for Blast Waveform Superposition

Based on the LSTM model, the motion state of the particles in the medium of the blasting wave over time is firstly studied and trained. After the training of the blasting wave, a set of parameters referring to the particle motion state is then obtained. Subsequently, the motion state of the monitoring point can be predicted. In addition, a complete data set of the media point motion state is considered, which includes the coordinates of the mass point, speed, acceleration, and latitude coordinates of each coordinate component $(x, y, z)$, and the time interval between sampling points. The input data are $\{x_1, x_2, \ldots, x_n\}$ for the media monitoring points at n consecutive equal intervals. For the sampling data that correspond to any time t, either the speed or acceleration is included. The output data are n consecutive equally spaced speed or acceleration $\{y_1, y_2, \ldots, y_n\}$ at the monitoring points.

$$x_t = \left\{ v_x, v_y, v_z \right\} \text{ or } x_t = \left\{ a_x, a_y, a_z \right\} \tag{1}$$

Then, the prediction model expression is

$$y_t = f(x_t) \tag{2}$$

According to the characteristics of the input and output data, the application scenario of this paper can be a multiple-to-multiple model, that is, each time the input has a sequence output (as shown in Figure 1).

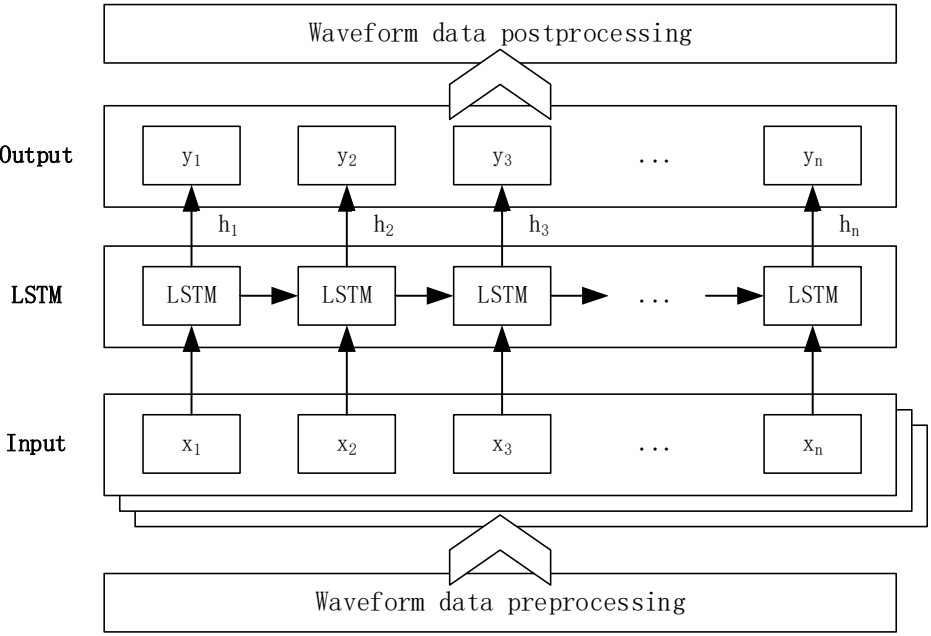

**Figure 1.** Framework of LSTM-based blasting waveform prediction model.

The model is divided into four layers: the first layer is the data pre-processing, including the noise reduction of the vibration waveform data and the slicing of the data. Due to the limited amount of monitored data, the slicing method can be used to increase the amount of data. The second layer is used to process the data into the input form required by the model, that is, normalization, matrixization, and 3-D tensor combination. In the LSTM constructed herein, the input is a 3-D tensor with the input tensor shape (time_steps = $n$, $n$_samples, dim_input). It also retains the traditional mini-batch gradient descent training method of a feed-forward network, but the difference is that a time-steps dimension is added, which indicates that each sequence contains blast vibration waveform data. The third layer is the LSTM layer, which utilizes the LSTM model to learn the data and train the model. The fourth layer is the output layer. The results that are output by the LSTM layer are normalized and decoded, and then the predicted waveform data are the output. The main frequency and energy distribution calculation can be undertaken based on the output waveform data.

### 2.2. Structure of the Memory Unit of LSTM

The simplified structure of the Recurrent Neural Network (RNN) is shown in Figure 2. The notable feature of the hidden layer design of the RNN model is a self-connected edge, which connects the hidden layers of the RNN at all times. The hidden layer value (s) will propagate along with the network (t). The value of the hidden layer at a certain time of the model is calculated from the output of the previous time and the input of the current time. Therefore, RNN has the capacity to learn time-series data.

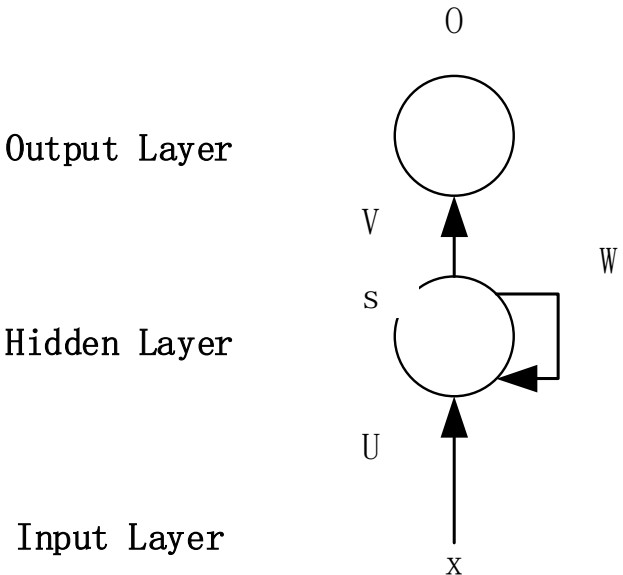

**Figure 2.** Simplified structure of the RNN.

After the traditional RNN model is separated, it can be equated to a multi-layer feed-forward neural network in the horizontal direction, that is, in the time dimension. The number of layers in the network is the length of the time-series data processed by the RNN. If the length of the data sequence is too long after being processed, the gradient disappears (explodes). Once these problems occur, the information remembered by the model is lost. In other words, the network will forget the earlier input. These problems are also the most important factors that limit the ability of an RNN to solve practical problems.

LSTM is an improved model of RNN. LSTM adds a state c to the RNN and uses it to save long-term states, thereby solving the problems of RNN models with only one hidden layer state h (Figure 3). For the calculation of the states c, h, and the model output, the LSTM adds three composite units for input calculation, such as an input gate, output gate, and forget gate. LSTM uses special layers to store and transmit information over a

long period. Using the multiple calculations of three gates, it overcomes the problem of vanishing gradients. This allows the LSTM model to avoid gradient disappearance when processing data with a long time span. In this way, it has good convergence.

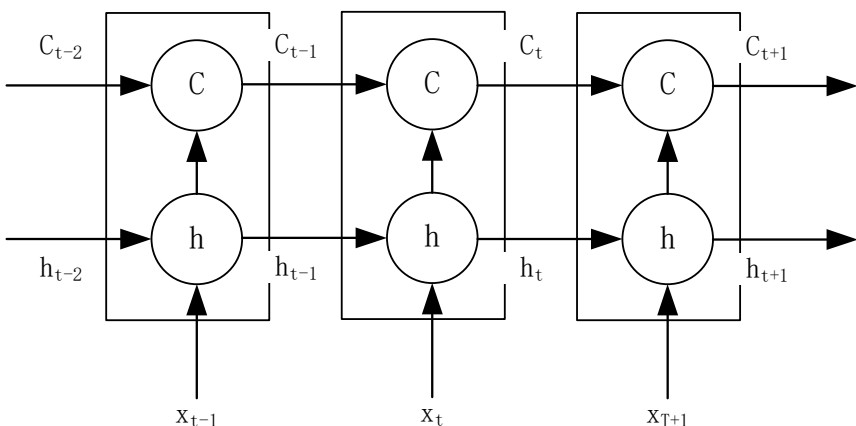

**Figure 3.** Schematic diagram of the internal structure of the LSTM.

The gate of the LSTM is a fully connected layer. The input is a vector *x*, which is calculated with the weight vector W and the bias term b, and then filtered by the activation function. The output will be in the form of a vector of real numbers from 0 to 1. The calculation equation is as follows:

$$g(x) = \delta(Wx + b) \tag{3}$$

The gate σ used in the LSTM hidden layer uses a sigmoid function (Figure 4) within a range of (0, 1). It multiplies the output of the gate by the input vector to determine how much information to pass. When the gate output is 0, the product result with the input vector is 0, and the information cannot pass. At this time, the gate is closed. When the output is 1, the input vector is unaffected and the gate is open. Due to the range characteristic of the sigmoid function, the σ-gate is always half-open. LSTM uses a forget gate and input gate to control unit state c. The forget gate determines how much of the unit state ct-1 at the previous time remains at the current time. The input gate determines how much of the network's input at that time is saved in the unit state. The LSTM uses output gates to control how much of the unit status (ct) is output to the current output of the LSTM.

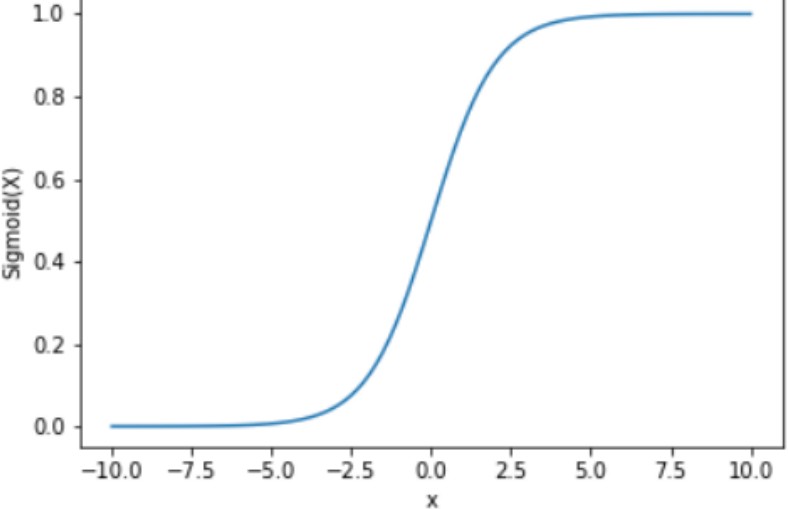

**Figure 4.** Graph of sigmoid function.

### 2.3. Acquisition of Blasting Vibration Waveform Data

When predicting the blasting vibration, the start time of the blasting vibration wave transmitted to the target point of each blast hole needs to be determined; it is necessary to obtain the field ground wave propagation waveform. In fact, the ground wave propagation process also reflects the site's geological conditions. The blasting vibration meter is used in this experiment and the sensor can monitor three channels (*x*, *y*, and *z*). The software provided by the vibrometer system completes the FFT report and time coordinate analysis, and it can also integrate, distinguish, convert, add, replace, filter, and intercept the waveform. The raw data obtained by the vibrometer are derived, and the proposed algorithm is then used for analysis and prediction (Figure 5).

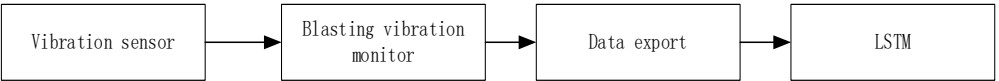

**Figure 5.** Schematic diagram of blasting vibration data acquisition process.

### 2.4. Training Algorithm

At present, the Adam algorithm and its derived optimization algorithms are commonly used in engineering applications to train neural networks. The batch-size selection of each batch for the training data is critical to the training process of the model. In general, the larger the batch-size setting, the more accurate the model's gradient descent direction, and the fewer batches needed to run a full data set (epoch). Larger matrix operations increase the concurrency rate and increase memory utilization. Correspondingly, if the batch size is set too large, it will cause too much memory consumption and the shortcomings of the weight update process will make it inefficient. In this consideration, when the batch size takes a certain value, the training effect of the model is the best, and the value is usually the power of *n*. Since this model is a waveform prediction model, it is classified as a multi-input, multi-output model. That is to say, the number of layers of the input layer is the number of features in the training data set. When the data contain three features of speed, acceleration, *x*, *y*, and *z*, the number of layers in the input layer is three; because the predicted value of the model output is the three-dimensional vector of the vibration velocity or acceleration of the particle, the number of output layer size is three. Different speed values may cause large differences in the predicted value, which will affect the accuracy of the model prediction. Therefore, the data of the input model need to be standardized (normalized).

## 3. Waveform Prediction Experiment: Porous Rock Blasting

### 3.1. Data Preparation

If the lithology near the blasting location is unchanged, the measured single-hole vibration waveform can be obtained through a single-hole blasting vibration test. The vibration of the porous blast at different locations is then superimposed upon the single-hole vibration waveform to obtain the peak vibration velocity of each observation position and realize the prediction of the vibration intensity of the porous blast. In the calculation process, each wavelet is superimposed in the order of delay and the distance from the measurement point. Blair proposed a full waveform, linear superposition model for this purpose [45]. It is assumed that each blast hole acts on the propagating medium, and that there is no effect between the blast holes. The model can be expressed as:

$$v(t) = \sum_{n=1}^{N} K d_n^{-B} w_n^A s_n(t - \delta_n) \tag{4}$$

where $v(t)$ is the blasting vibration velocity of the monitoring point at time $t$, $N$ is the number of blast holes in this area, $K$ is the attenuation coefficient, $B$ is the attenuation coefficient of the distance between the blast hole and the monitoring point, $A$ is the blast hole charge coefficient, $d_n$ is the distance from the nth blast hole to the test point, $S_n(t)$

is a function of the velocity waveform of a single initiation of the nth blast hole, and $\delta_n$ is the time taken for the vibration caused by the nth blast hole to be transmitted to the monitoring point.

For the convenience of calculation, as shown in Figure 6, there are five blast holes, each of which has the same charge, $w_n = 1$. The five blast holes detonate at the same time, so that the waveform excited by the blast holes are the same and are formed by superimposing two sine waves. Assuming $K = 1$, $B = 1$, and $A = 1$, forty randomly generated pairs of sine waves are superimposed into a vibration waveform $S_n(t)$ of the 20 blast holes. Thus, the waveform of the two points $P_a$ and $P_b$ on the *X*-axis is calculated as test data.

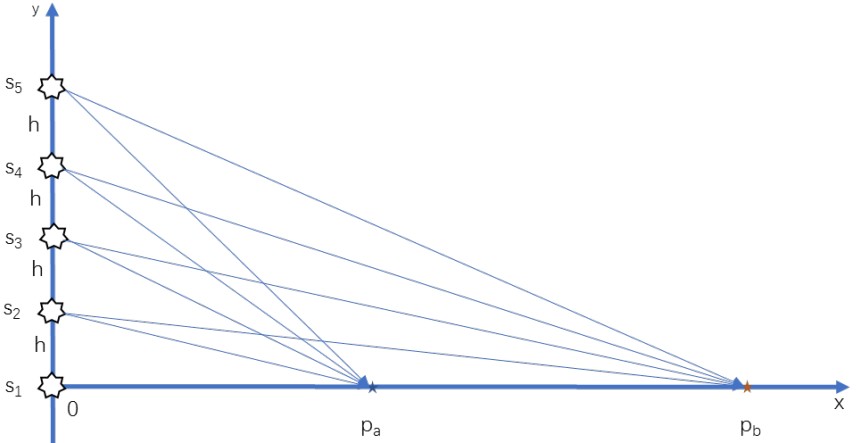

**Figure 6.** Geometrical relation between the blast hole and the measuring point.

### 3.2. Evaluation of the Prediction Result

The prediction data generated by the LSTM-based waveform prediction model is the three-dimensional data (*x*, *y*, *z*) for denoting the waveform. Therefore, this experiment uses Euclidean distance and mean square error, which are widely used in multi-dimensional prediction models as the evaluation function of the prediction accuracy of the model. There are three evaluation indicators, which are the average Euclidean distance, maximum Euclidean distance, and mean square error. The calculation formulas for the three evaluation indicators are as follows:

$$MSE = \frac{1}{N}\sum_{t=1}^{N}(y_t - \hat{y}_t)^2 \tag{5}$$

$$Euclidean\ Distance = \sqrt{\sum_{1}^{N}(y_t - \hat{y}_t)^2} \tag{6}$$

$$Max\ E\_Distance = \max(Euclidean\ Distance) \tag{7}$$

where $\hat{y}_t$ is the predicted value output by the model at time *t*, $y_t$ is the true value corresponding to the trajectory data at time *t*, and *N* is the number of samples in the prediction set.

### 3.3. Determination of LSTM Network

The simulation data set used in this experiment contains 20 sets of sampling data. The training set is randomly divided into training set prediction sets, and then the 20 sets of data are randomly sorted. The process is iterated 3000 times. Through the process of random sample division and multiple experiments using randomly divided samples, more randomness is introduced to obtain a more stable model.

### 3.4. Experimental Result

The test system of this experiment includes: Intel$^{(R)}$ Xeon$^{(R)}$ Bronze 3106 processor with a main frequency of 1.7 GHz, 16 GB of running memory (RAM), NVIDIA GeForce

RTX 2080 Ti graphics card, and 64-bit Microsoft Win10 professional operating system. The system includes data simulation and model construction. The programming language used in this experiment is Python 3.7.1 (64-bit), and the experimental operating environment is Spyder 3.3.6. During the construction of the LSTM-based vibration wave prediction model, multiple libraries were used for data processing and matrix operations between model nodes. The TensorFlow library version is 1.14.0, the Numpy library version is 1.16.5, and the Matplotlib library version is 2.2.2.

Figure 7 shows the curve of the vibration velocity of particle Pb as a function of the time series (prediction series). Among them, the vertical axis and the horizontal axis represent the vibration velocity on the corresponding time series. There are 2000 data points in total, matching the data volume of the test set. The orange curve is the vibration speed predicted by the model, and the blue curve is the real vibration speed at the corresponding point $P_b$. Starting from the 0.05 s prediction point, the predicted trajectory curve and the real trajectory curve are in good agreement, as shown in Figure 6.

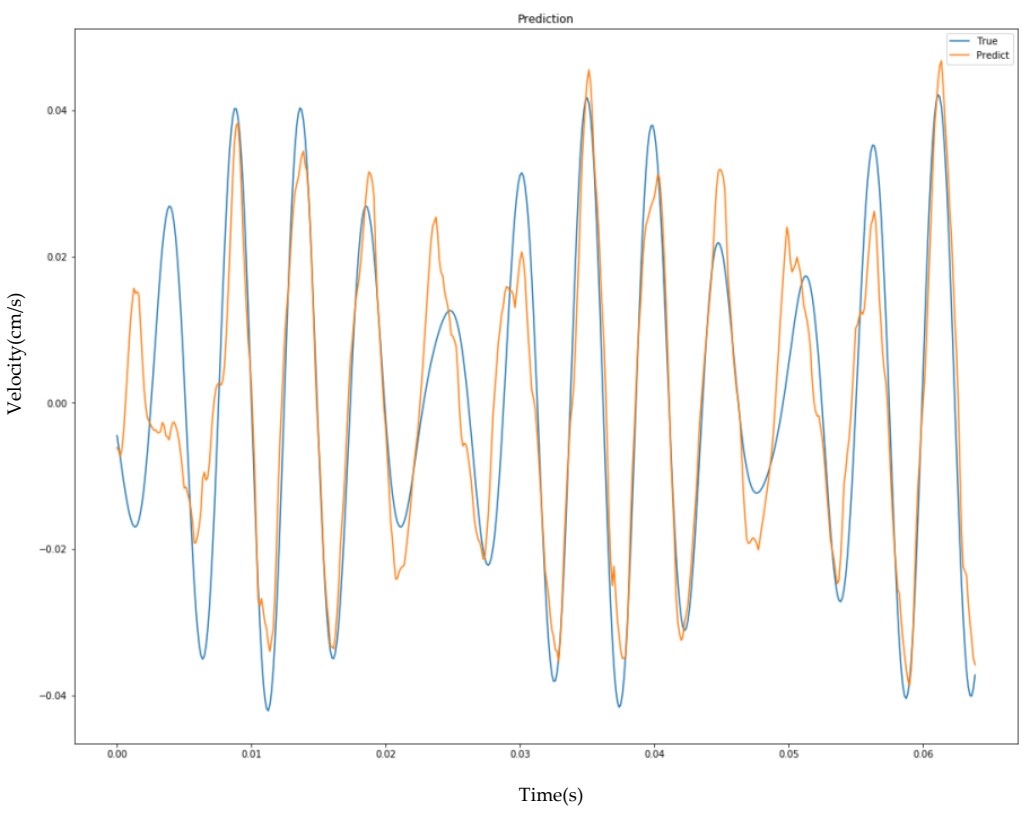

**Figure 7.** Comparison between the predicted and real vibration velocity waveform.

Figure 8 shows the curve of the loss function of the prediction model training process with the number of iterations of the inverse optimization of the training process. The curve denotes the change process of the mean square error MSE of the loss function during the training process, and the horizontal axis is the number of reverse iterations of the model training process.

After the 1500th iteration, the curve is quasi-stable, which indicates that the model can be deemed stable thereafter, and the error of the loss function is about 0.01. From the perspective of convergence speed, the fluctuations are mainly concentrated in the first 1000 iterations. Due to the Adam adaptive optimization algorithm, the learning speed of the model can be dynamically adjusted.

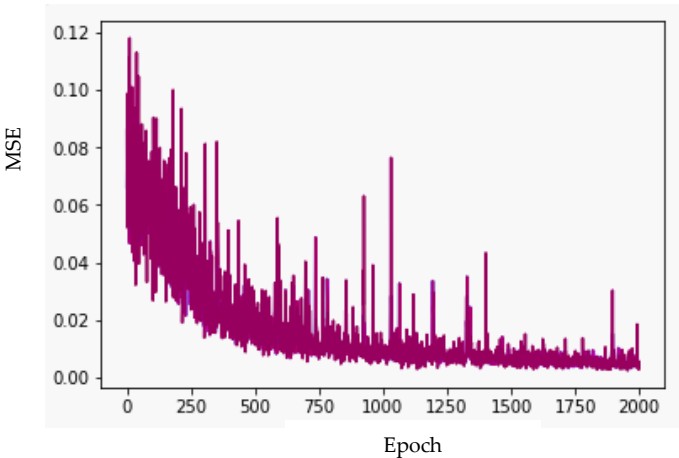

**Figure 8.** Loss function curves (in training).

## 4. Field Application

### 4.1. Engineering Background Overview

Hongtuoshan Copper Mine is located in Hongtuoshan Town, Qingyuan County, Fushun City, Liaoning Province (Figure 9). The mine has been operated since 1958 and has a mining history of more than 50 years. At present, the mining depth is more than 1200 m, and 21 middle sections were mined. The Hongtuoshan deposit is located in the upper wall of the Hunhe large fault zone and belongs to the pyrite-type, vein-shaped copper–zinc deposit. It is mainly composed of biotite plagiogneiss and amphibole gneiss. The primary joints and fractures of the rock mass in the mining area are undeveloped, and the geological structure is simple. The ore rock has a hard texture, compact structure, small porosity, and low water content. The ore solidity coefficient is 8–10, the surrounding rock solidity coefficient is 10–14, the uniaxial compressive strength of ore is about 70 MPa, the ore integrity coefficient is 0.88, and the RQD value of the rock mass exceeds 65%.

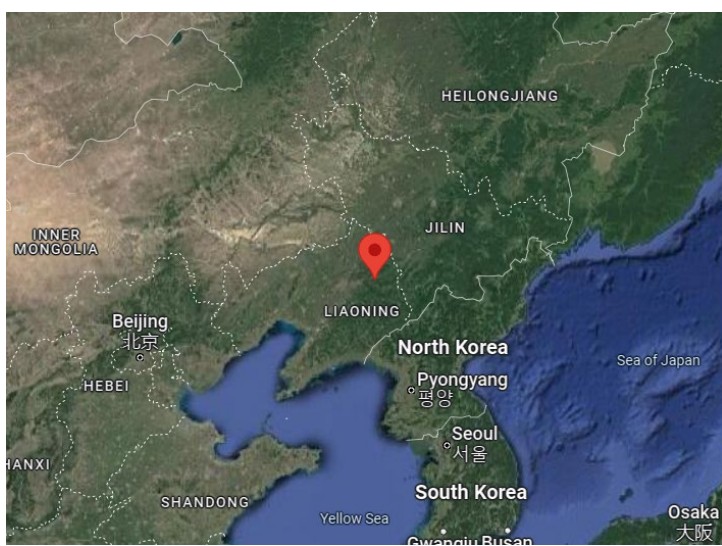

**Figure 9.** Mine location for blasting engineering test.

### 4.2. Blasting Parameters

The 47 Stope in the middle of −707 level is located in the No. 3 vein of the Hongtuoshan deposit. The ore body goes northeast between 70° and 75° and tends to tilt South. The specific blasting parameters of the stope are as follows: the blasthole diameter is 75 mm. The mesh parameters are 2 m × 1.2 m. The hole depth ranges from 7.4 to 12.8 m, and

the packing length is 1 m. The explosive is a No. 2 rock emulsified explosive (Explosion velocity: 4000~4300 m/s, Density: 1.10 kg/cm³, Detonation distance: 7 cm). The specific explosive consumption is 0.255 to 0.301 kg t − 1. The blast area division and blasting sequence (I–VII) are shown in Figure 10.

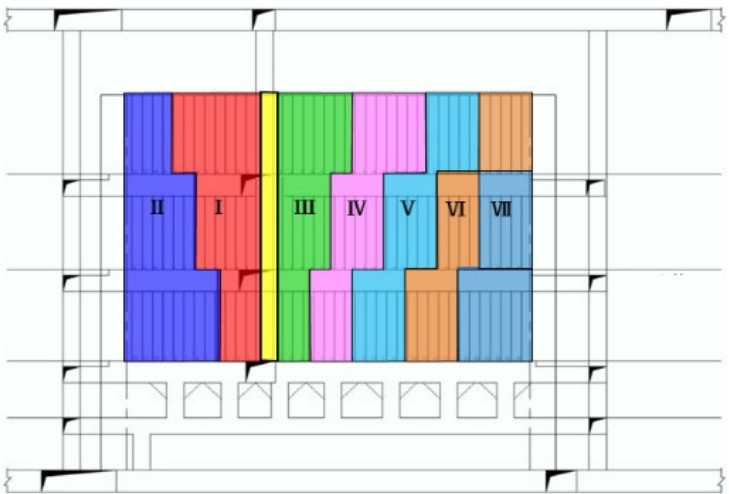

**Figure 10.** Division of blasting area and the blasting sequence.

### 4.3. Blasting Vibration Monitoring Scheme

According to the actual production conditions and measuring point conditions at the site, blasting vibration tests were carried out in the middle section of Hongtuoshan Mine-707. The schematic diagram of the location of the measuring point and the explosion zone is shown in Figure 11. The measuring points are designed along the floor area of the transportation lane, ensuring they were in a straight line. A total of nine points(①–⑨) were selected as candidate measurement points, and five (①–⑤) were selected as actual measurement points according to the location of each blasting area.

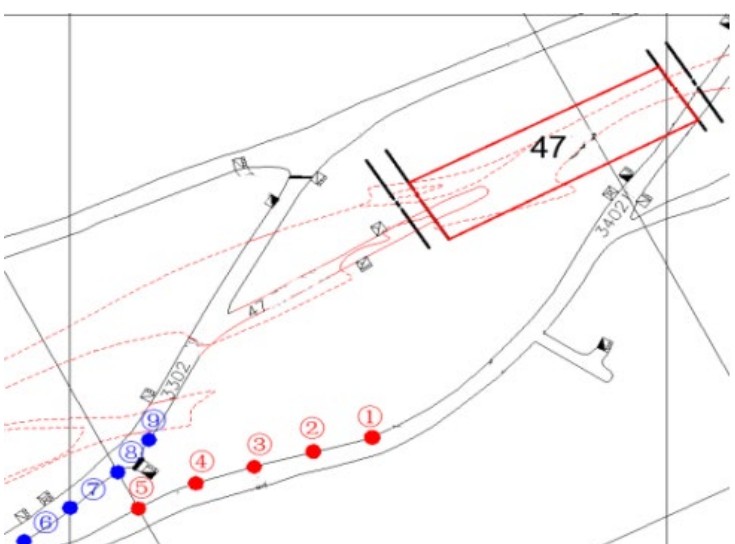

**Figure 11.** Layout of measuring point and the locations of blasting point.

The equipment used for monitoring was a CBSD-VM-M01 network vibration meter, which includes: one control analyzer, six smart sensors, one 3 G router, and an instrument box. Among them, the network control analyzer function module included equipment detection, parameter setting, task writing, data reading, equipment control, real-time monitoring, on-site photography, and coordinate positioning. Device detection was mainly

used to detect smart sensors. The parameter setting was employed to set test parameters for smart sensors on the spot. Task writing described the task of vibration measurement and completion of the related vibration measurement record form. Data were recorded by the smart sensor. Device control was used to control smart sensors. Real-time monitoring was the real-time vibration test operation. Live photography was used to take photographs and upload them in real time. Coordinate positioning was used to check the coordinates of test points and blasting points. Six smart sensors were used to receive the blasting vibration signal, including the vertical, horizontal tangential, horizontal, and radial vibration speeds of the particle vibration and the main vibration frequency. Its performance parameters are as follows: measurable frequency range: 5–5000 Hz, maximum range 35.5 cm/s, minimum range 0.01 cm/s; recording time: 1~999 s; number of channels: XYZ three channels; maximum allowable number of events: 5000.

### 4.4. Blasting Vibration Monitoring Data

Five on-site blasting vibration tests were carried out. Although the five blasting tests were all carried out in the same stope, the specific blasting locations were different. In addition, according to the requirements of mining work, the charge amount of these five blasts is also different. Vibration testers can be used to obtain vibration waves in three directions: horizontal radial (X-direction), horizontal tangential (Y-direction), and vertical (Z-direction). The parameters of the example were measured at No. 5 measuring point during the sixth blasting operation. The specific parameters are listed in Table 1, and the resulting waveform diagram is illustrated in Figure 12.

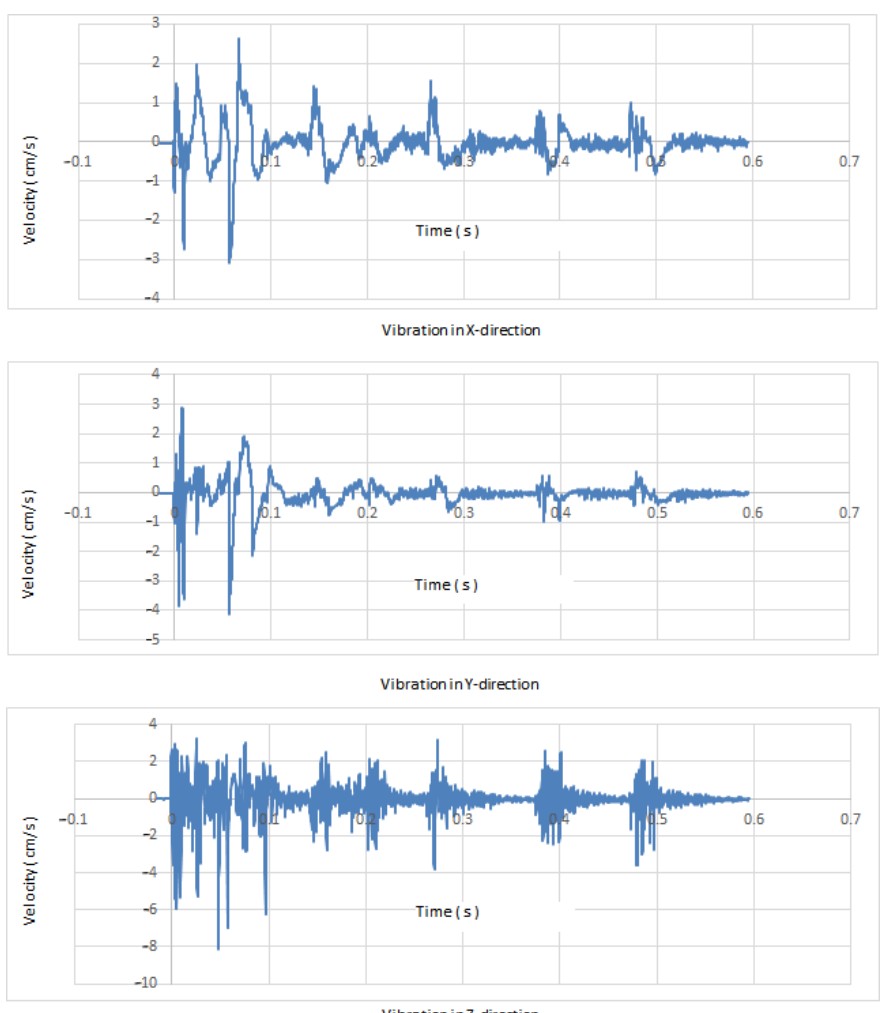

**Figure 12.** The vibration waveform of the fifth blasting.

**Table 1.** Monitoring data of the sixth blasting vibration monitoring point.

| Channel Number | Maximum Velocity (cm/s) | Maximum Moment (ms) | Vibration Duration (s) | Dominate Frequency (Hz) |
|---|---|---|---|---|
| X-direction vibration | 3.06 | 20 | 0.539 | 285 |
| Y-direction vibration | 4.11 | 20.4 | 0.518 | 243.726 |
| Z-direction vibration | 8.07 | 62.2 | 0.654 | 305.62 |

### 4.5. Results and Discussion

The Keras framework was used to implement the LSTM prediction model. The MSRA method was used for the weight initialization. The ReLU function was selected as the hidden layer activation function, and the Softmax was used as the output layer activation function. DropOut can prevent the model from overfitting and improve the performance of the model (according to experience, it is set to 0.5). 47 Stope was subjected to multi-stage blasting with the same row and the same stage, and the difference between the rows lies solely in the blasting. The artillery was divided into a minimum of 9 sections and a maximum of 12 sections. In the blasting process, the charge of each stage is different. The more accurately to find the relationship between the charge and the vibration speed of the particle and the frequency of the main vibration, signals with obvious segmentary characteristics were divided accordingly to find the delay time. The waveform generated by each micro-blasting event was regarded as an independent blasting operation. The entire dataset was fragmented to form 15 sets of data. The 14 sets of data were used for model training and the 7th data was used for prediction testing. The experimental results are shown in Figure 13.

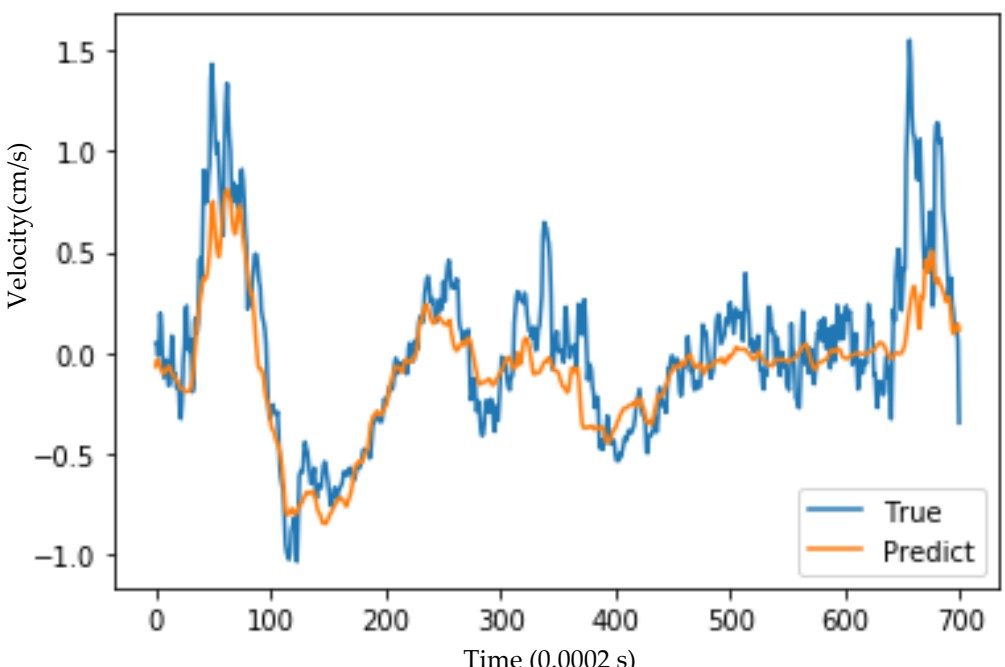

**Figure 13.** Comparison between the predicted and true field vibration velocity waveform.

Figure 13 demonstrates the curve of the vibration velocity of the particle with time series (prediction series). Among them, the vertical axis and the horizontal axis represent the vibration velocity on the corresponding time series, and there are 700 data points in total. The red curve denotes the vibration speed predicted by the model, and the blue curve denotes the real vibration speed at the monitoring point at the corresponding time: the predicted seismic velocity curve basically agreed well with the actual seismic velocity curve in terms of both trend and value.

The main factors affecting LSTM prediction accuracy are sequence length, training algorithm, and a number of hidden layer nodes. To obtain appropriate parameters, different sequence lengths, a number of hidden layer nodes, and training algorithms are studied. The optimal parameters are obtained according to the model loss functions under different conditions.

For the sequence length, in the experiment, the number of iterations is uniformly set to 4000, and the remaining parameters are kept unchanged. The model is trained by increasing and decreasing the sequence length (Figure 14). It is found that the sample size has a significant effect on the convergence speed of the model. For cases in which the sequence length is too short, when the sequence length is less than 500, the deep neural network model cannot fully learn the hidden rules in the time series, which leads to a decrease in model accuracy. For cases in which the sequence length is too long, when it is greater than 700, the accuracy of the model will not be improved, and the gradient explosion tends to occur, and the training time of the model will be greatly increased.

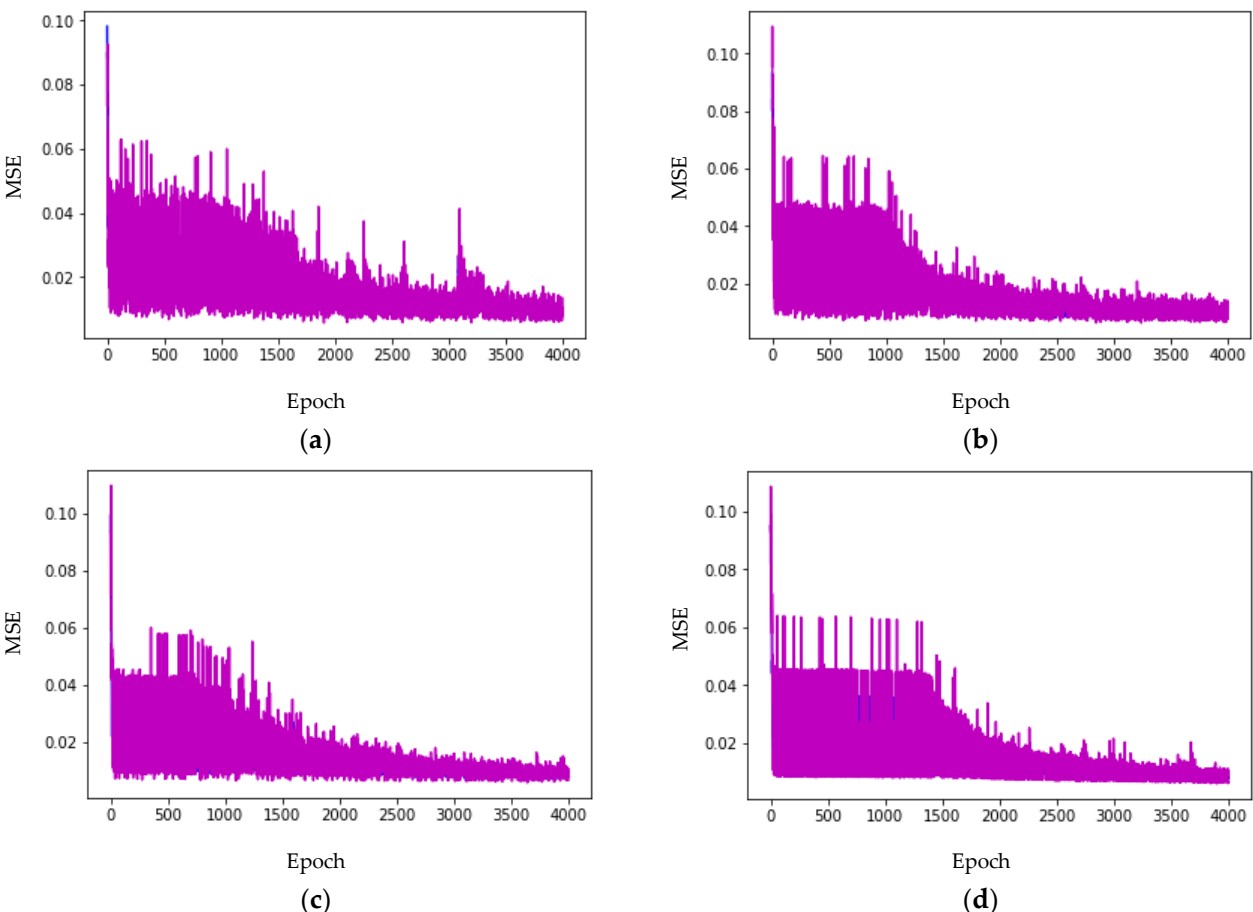

**Figure 14.** Loss function curves for a sequence length of (**a**) 400, (**b**) 500, (**c**) 700, and (**d**) 900.

In terms of model structure, the choice of the number in the hidden layer has a significant effect on the performance of the model. When the number of nodes in the hidden layer is inappropriate, "overfitting" readily will occur. At present, there is no method available for determining the number of nodes in the hidden layer. Most researchers have quoted the empirical values and experimental results to determine the number of nodes in the hidden layer.

In this study, the number of hidden layers is initialized to 10, and the number of hidden layers is gradually increased by 10 digits. Based on the model loss function of the model on the test set sample, it is concluded that the loss function changes with the increase of the number of hidden layers. The specific analysis results are shown in Figure 15.

The models with different numbers of hidden layers are trained and the training results are compared and analyzed. If the number of hidden layers is too small, the accuracy of the model will be reduced to less than 20, the accuracy of the model validation set and the training set will be reduced, and the model will underfit the data. When the number of layers exceeds 40, the accuracy of the model training set increases, but the accuracy of the validation set decreases, the model is in an over-fitting state, and the training time becomes too long.

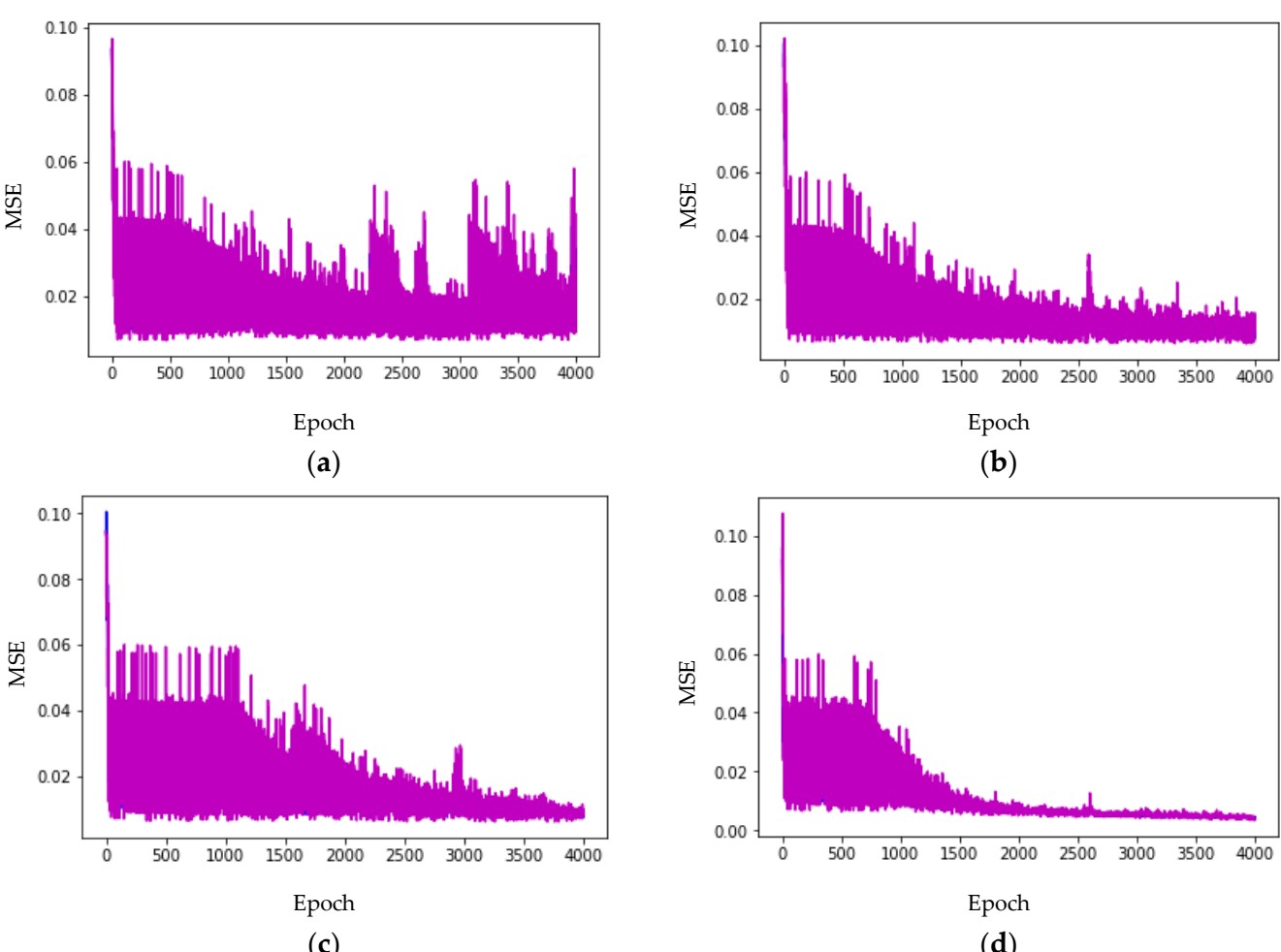

**Figure 15.** Loss function curves when the number of hidden layers is (**a**) 10, (**b**) 20, (**c**) 30, and (**d**) 40.

For optimization methods, the RMSprop and Adam algorithms are more suitable for optimizing recurrent neural networks and LSTMs. To compare the effect of the optimization algorithm on the model, four algorithms are used herein. In this experiment, the number of nodes in the hidden layer is 30, and the other parameters are set to the aforementioned uniform values. Figure 16 shows the changes in the loss as the number of iterations increases when the RMSprop, Adam, Adamax, and Adagrad optimization algorithms are used. According to the experimental results, the model using Adam's optimization method offers a better algorithm performance than the RMSprop model in terms of prediction accuracy and model convergence speed. Adamax and Adagrad have not achieved relatively stable convergence in this model. The Adam algorithm is more stable in the validation set; because Adam inherits the advantages of the RMSprop algorithm, it does not need to manually set the learning rate, does not consume memory, and is suitable for solving large data sets and high-dimensional space problems: it is a popular optimization algorithm and is thus selected here to train the model.

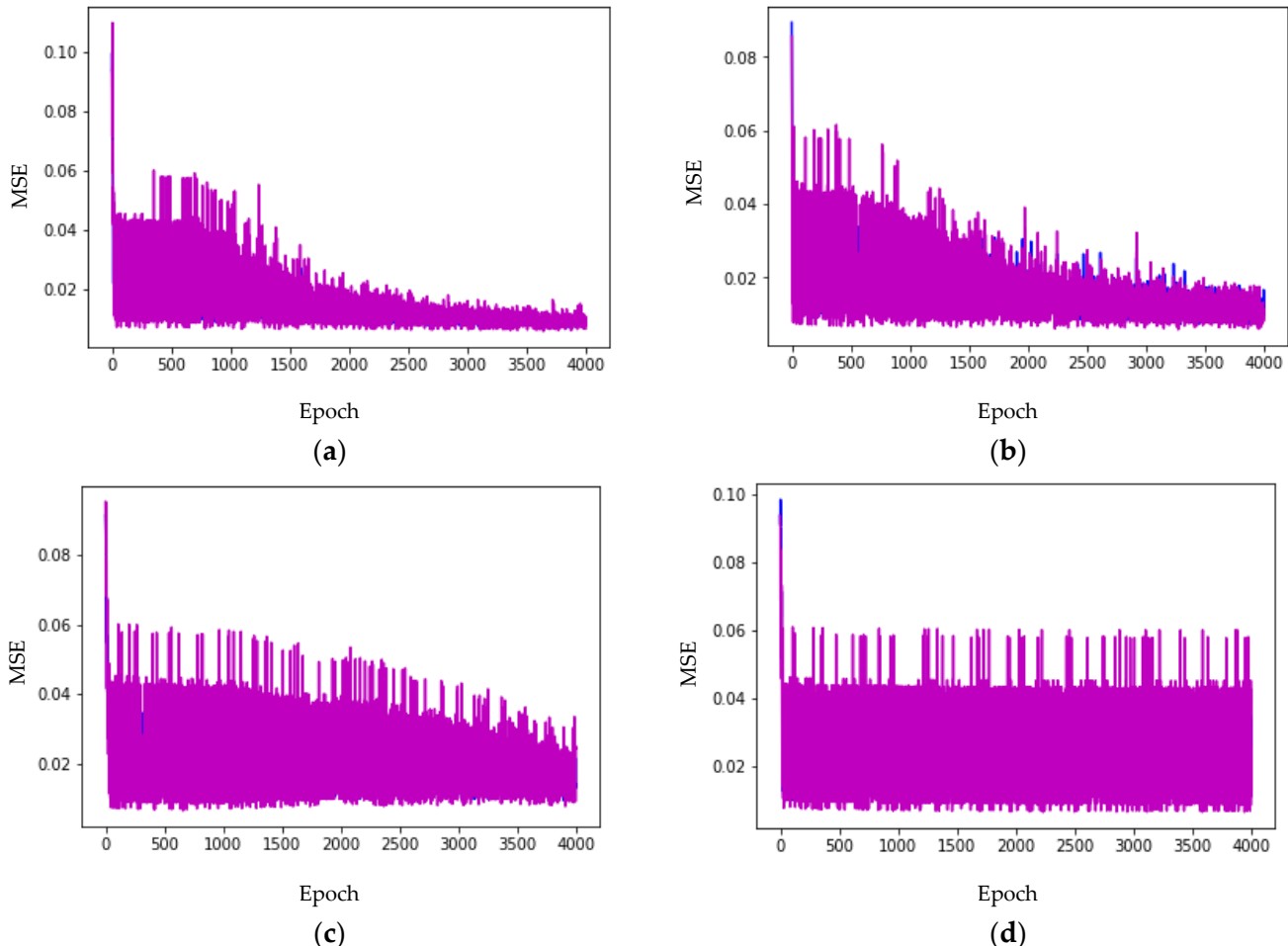

**Figure 16.** Loss function curves when the optimisation function is (**a**) Adam, (**b**) RMSprop, (**c**) Adamax, and (**d**) Adagrad.

In summary, the effects of different sequence lengths, training functions, and a number of hidden layer nodes on model accuracy are analyzed experimentally. Considering the training time and the accuracy of the model, the sequence length is set to 700, the Adam optimisation algorithm is used, and the hidden layer contains 30 nodes, which is more suitable for training the model proposed in this paper.

## 5. Conclusions

Based on the LSTM deep neural network, a prediction model of a full waveform of blasting vibration is proposed in this study. The established model is trained and tested, and experiments are performed to verify the efficiency of the proposed prediction model. According to the experimental results, the proposed LSTM network method for vibration full waveform prediction is reliable. The full-waveform prediction analysis is not limited to the peak vibration velocity but extends to the waveform of blasting vibrations, so that the frequency and duration of blasting vibration are included in the vibration analysis and evaluation, and the prediction result is more comprehensive.

Referring to the sequence length, when the sequence length is too short, the deep neural network model cannot fully learn the hidden rules in the time series, which leads to a decrease in model accuracy. When the length of the sequence is too long, the accuracy of the model will not be improved, and the gradient explosion tends to occur, and the training time of the model will be greatly increased.

In terms of model structure, models with different numbers of hidden layers are trained and the training results are compared and analyzed. If the number of hidden layers

is too small, the accuracy of the model will be reduced. If the number of layers is too small, the accuracy of the model validation set and the training set will be low, and the model will be underfitted. With too many layers, the accuracy of the model training set increases, while the accuracy of the validation set decreases and the model is in an overfitted state.

In terms of optimization methods, RMSprop, Adam, Adamax, and Adagrad optimization methods are used to optimize the network training process. The model using the Adam optimization method is superior to the model using RMSprop optimization in terms of prediction accuracy and model convergence speed. Therefore, for the sake of the present research, the proposed prediction model is based on the Adam method.

Although the LSTM deep neural network is effective for the prediction of the blasting vibration full waveform, the current prediction accuracy rate still needs to be improved. This research focussed on using the vibration waveforms of known monitoring points to predict the full waveform of unknown points. In the future, the author will try to consider the relationship between the characteristics of the blasting source and the spatial position of the monitoring point and establish a model using the blasting source waveform to reduce dependence on the monitoring point.

**Author Contributions:** Data curation, G.Z.; Project administration, F.Z.; Supervision, Y.W.; Writing—draft, Y.W. and Y.L. All authors have read and agreed to the published version of the manuscript.

**Funding:** This work was funded by National Key R&D Programme of China (2021YFC3001304) and the Fundamental Research Funds for the Central Universities (N2101044).

**Data Availability Statement:** Not applicable.

**Conflicts of Interest:** The authors declare that the research was conducted in the absence of any commercial or financial relationships that could be construed as a potential conflict of interest.

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
