# Peer review of "Full Waveform Prediction of Blasting Vibration Using Deep Learning"

_sustainability, doi:10.3390/su14138200_

Round 1
Reviewer 1 Report
The prediction of blast vibration is a relatively challenging task. In engineering applications, it is usually calculated or predicted using the formula method.
The authors use the LSTM method for full waveform prediction, which is a good attempt of deep learning methods in blast vibration prediction. In mine production, the location of blasting is constantly moving as the mining work continues, so the spatial physical properties of the ore rocks in the study area such as fractures are constantly changing.
It is hoped that the authors will try to apply the model trained in this exercise directly to more distant blast sites in future studies and try to analyze the predicted effects. Further research will be done to increase the applicability of this method in various rock masses.
Reviewer 2 Report
Dear authors,
The topic is attractive and has high research potential. Vibrations caused by blasting are significant nowadays, as there are many blastings in urban areas.
Below you can find a few comments suggested for better understanding and improvement of article quality:
Line 174: The gate σ used in the LSTM hidden layer uses a sigmoid function within a range of (0, 1)- Can you please add a figure that represents the sigmoid function and/or equation
Line 310: The explosive is a No. 2 rock emulsified explosive- Please provide more data about the explosive (detonation velocity, density …etc.)
Line 325: The equipment used for monitoring was a CBSD-VM-M01 network vibration meter – Please provide more data about the CBSD-VM-M01- for example (response standard, frequency range, velocity range and resolution)
Line 356-357: In the blasting process, the charge of each stage is different. The more accurately to find the relationship between the charge and the vibration speed of the particle and the frequency of the main vibration. - I believe you considered charge per delay; it has to be stated in the article to avoid possible misunderstanding.
Figure 12: please add units on x and y axis
Figures 6,7, 13,14 and 15 – same as Fig 12
Line 369-371: The red curve denotes the vibration speed predicted by the model, and the blue curve denotes the real vibration speed at the monitoring point at the corresponding time the predicted seismic velocity curve basically agreed well with the actual seismic velocity curve in terms of both trend and value. The curve is orange in Fig 12. Please explain big difference between the true and predicted curve at the end of the curve (last peak).
Some minor typing mistakes noticed:
Line 12: modelis
Line 13: study. To (It seems that one space tab too much)
Line 36: Blasting
Line 229: nth
Line 389: sequence length (It appears that one space tab too much)
Please check the whole text once again and correct possible mistakes.
